# Postoperative Outcomes of Pre-Pectoral Versus Sub-Pectoral Implant Immediate Breast Reconstruction

**DOI:** 10.3390/cancers16061129

**Published:** 2024-03-12

**Authors:** Gilles Houvenaeghel, Marie Bannier, Catherine Bouteille, Camille Tallet, Laura Sabiani, Axelle Charavil, Arthur Bertrand, Aurore Van Troy, Max Buttarelli, Charlène Teyssandier, Agnès Tallet, Alexandre de Nonneville, Monique Cohen

**Affiliations:** 1Aix-Marseille University, CNRS (National Center of Scientific Research), INSERM (National Institute of Health and Medical Research), Paoli-Calmettes Institute, Department of Surgical Oncology, CRCM (Research Cancer Centre of Marseille), 13009 Marseille, France; 2Paoli-Calmettes Institute, Department of Surgical Oncology, CRCM (Research Cancer Centre of Marseille), 13009 Marseille, France; bannierm@ipc.unicancer.fr (M.B.); laurentc2@ipc.unicancer.fr (C.B.); sabianil@ipc.unicancer.fr (L.S.); charavila@ipc.unicancer.fr (A.C.); bertranda@ipc.unicancer.fr (A.B.); vantroya@ipc.unicancer.fr (A.V.T.); buttarellim@ipc.unicancer.fr (M.B.); cohenm@ipc.unicancer.fr (M.C.); 3Paoli-Calmettes Institute, Department of Radiotherapy, CRCM (Research Cancer Centre of Marseille), 13009 Marseille, France; camille.tallet@etu.univ-amu.fr (C.T.); charlene.teyssandier@etu.univ-amu.fr (C.T.); richarda@ipc.unicancer.fr (A.T.); 4Aix-Marseille University, CNRS (National Center of Scientific Research), INSERM (National Institute of Health and Medical Research), Paoli-Calmettes Institute, Department of Medical Oncology, CRCM (Research Cancer Centre of Marseille), 13009 Marseille, France; tassindenonnevillea@ipc.unicancer.fr

**Keywords:** breast cancer, reconstruction, prepectoral implant, post-surgical outcome, satisfaction, cost

## Abstract

**Simple Summary:**

A rapid evolution of IBR techniques has been reported, including prepectoral implant immediate breast reconstruction (IBR) with a mesh. In a monocentric cohort, subpectoral implant IBR was performed in 529 mastectomies (62.0%) and prepectoral implant IBR in 324 (38.0%), with a significant increase in prepectoral placement in recent years. Mesh was used in 176 prepectoral placements (54.3%). Any grade of complication was reported in 147 mastectomies (17.2%), with a significantly higher rate for prepectoral implant IBR, with no significant difference for grade 2–3 complications. Regression analysis showed that prepectoral implant was not significantly associated with any grade of complication, or with grade 2–3 complications. Prepectoral implant-IBR was associated with significantly shorter operative times. Costs above the median were significantly associated with subpectoral placement and mesh use. Prepectoral implantation can be considered a good and safe technique. However, patient selection may be necessary and we propose a complication risk score to aid decision-making.

**Abstract:**

Introduction: Immediate breast reconstruction (IBR) techniques are rapidly evolving. We compared the results from a single-center implant IBR cohort between subpectoral and prepectoral implants with and without a mesh. Methods: We analyzed all complications and grade 2–3 complications, the implant loss rate, the surgery time, the length of stay (LOS), patient satisfaction, the interval time to adjuvant therapy and cost, with a comparison between subpectoral and prepectoral implant IBR. Results: Subpectoral implant IBR was carried out in 529 mastectomies (62.0%) and prepectoral in 324, with a significant increase in prepectoral placement in recent years. Mesh was used in 176 prepectoral placements (54.3%). Any grade of complication was reported in 147 mastectomies (17.2%), with a significantly higher rate for prepectoral implant IBR (*p* = 0.036). Regression analysis showed that prepectoral implant was not significantly associated with any grade of complication or with grade 2–3 complications. Prepectoral implant IBR was associated with a significantly shorter operative time and lower LOS. Grade 2–3 complications were significantly associated with lower satisfaction. Higher costs were significantly associated with the subpectoral placement and mesh. A complication rate predictive score identified five groups with a significant increase in grade 2–3 complications. Conclusions: Prepectoral-M-IBR increased over time with no difference in complication rates compared to subpectoral-M-IBR. Prepectoral implant placement can be considered a safe technique.

## 1. Introduction

Breast cancer (BC) is the most common cancer and is the leading cause of cancer death in women, with 2.26 million new cases and 685.000 deaths expected in 2020 [1]. Although oncoplastic surgery has expanded the options for breast conservative surgery [2,3,4,5], mastectomy remains a common surgical option, ranging from 12 to 40%, as reported in the literature [6,7,8,9,10,11].

Many studies have reported the beneficial effects of immediate breast reconstruction (IBR) on patients’ quality of life without compromising oncological outcomes or time to adjuvant therapies. As a result, the incidence of IBR has increased in recent decades. It was reported to be 9.6% in China in 2018 [12], 14% between 2011 and 2016 in France (INCa report), and increased from 10% at the beginning of the 2000s to 23.3% in the year 2014 in the UK [13]. In our cancer center, the rate assessed between 2008 and 2014 was 16.1% (similar to that reported at the French level) and increased to 40.5% between 2016 and 2020 [14]. A large multicenter French cohort confirmed this trend, with a multivariate analysis showing odd ratios (ORs) of 2 between 2007 and 2009, and 2.5 between 2010 and 2019 compared with the previous periods 1999–2003 and 2004–2006 [15,16]. In the United States, breast reconstruction rates (IBR and delayed breast reconstruction) were 45% in 2010 and 54% in 2015.

Implant-based mastectomy IBR (implant-M-IBR) has long been the most common procedure [12,13,17,18]. However, in recent years, nipple-sparing mastectomy (NSM) has been increasingly used for both prophylactic mastectomy [19], primary BC [20,21,22,23,24] and local recurrence [25], offering better aesthetic results than skin-sparing mastectomy (SSM) [26,27,28,29,30,31,32]. IBR techniques rapidly evolved with the advent of prepectoral implant IBR [33,34], with or without a mesh, and robotic mastectomy IBR [35,36,37,38]. In addition, different types of mesh, acellular dermal matrices, synthetic absorbable matrices or non-absorbable matrices have been used [39,40,41,42,43].

The aim of this study was to report the results of a large single-center cohort of implant M-IBR in terms of postoperative complications, patient satisfaction and costs according to subpectoral or prepectoral implants with or without meshes. A predictive score for postoperative complications was established.

## 2. Materials and Methods

All implant M-IBR between January 2019 and November 2023 were prospectively included in the institutional database (study: M-IBR-PPRP-IPC 2022–014) and we retrospectively analyzed the data. The main prospectively recorded characteristics were as follows: age, year of surgery, reason for mastectomy, type of mastectomy, use of mesh (resorbable synthetic mesh TIGR Matrix^®^ Novus Scientific, Uppsala, Sweden), implant type, ASA status (American Society of Anesthesiologists), body mass index (BMI), smoking status, diabetes, previous surgery, previous radiotherapy, neoadjuvant chemotherapy (NAC), breast cup size, implant volume, mastectomy weight, surgical incisions, axillary surgery, adjuvant therapies and surgeons.

We analyzed the total postoperative complication rate and grade 2–3 complication according to the Clavien Dindo classification [44] (occurring within 90 days of surgery), complication type, re-operation rate, implant loss rate, duration of surgery (operative time between skin incision and skin closure), length of postoperative stay (LOS) (from day of surgery to discharge), patient satisfaction (very good, good, medium, bad and failure), time to adjuvant therapy and costs. For patients with bilateral mastectomies, analyses were performed on two procedures, and the duration of surgery was halved. Per-operative antimicrobial prophylaxis was systematically given to all patients with implant M-IBR.

The results were compared between subpectoral and prepectoral IBR in univariate and multivariate analyses. The choice of implant position (subpectoral or prepectoral) and the use of a mesh was at the surgeon’s discretion. If the mastectomy compartment was very large, the prepectoral implant was held in place by absorbable sutures between the outer edge of the pectoralis major muscle and the outer subcutaneous tissue, without the use of a mesh. Locoregional anesthesia with a pectoral block was systematically performed.

The cost of the initial procedure was assessed by adding the cost of the implant (400 Euros), the number of hospital days (1495.69 Euros per day), the operating room time (402.54 Euros per hour) and the mesh (1390 Euros for 20 × 30 cm). The operating room occupancy time was determined by the duration of surgery and 90 min for patient set-up, anesthesia, local anesthesia and awakening from anesthesia.

### Statistics

Quantitative criteria were analyzed using median, mean and 95% confidence interval (CI). Comparisons were determined using the Chi-2 test for qualitative criteria and the *t*-test for quantitative criteria. Factors significantly associated with the criteria analyzed were determined by a binary logistic regression adjusted for significant variables identified using univariate analysis. For binary logistic regression, quantitative criteria were divided into several categories: mastectomy weight > or ≤300 gr, BMI ≤ 24.9, 25 to 29.9, ≥30. An odds ratio (OR) with a 95% CI was used as the effective measure.

We calculated predictive scores for [1] any complication and [2] grade 2–3 complications using the ORs of significant factors derived from the logistic regression. The performance of these scores was analyzed by calculating the area under the curve (AUC). Statistical significance was set at *p* ≤ 0.05. Analyses were performed using SPSS version 16.0 (SPSS Inc., Chicago, IL, USA).

## 3. Results

From January 2019 to November 2023, 2002 mastectomies were performed including 900 IBR (44.96%): 853 implant-M-IBR (42.6%) and 47 other procedures M-IBR. The implant M-IBR rates were 47.9%, 29.4%, 40.0%, 44.5% and 55.1% in the years 2019, 2020, 2021, 2022 and 2023, respectively.

Subpectoral implant M-IBR was performed in 529 mastectomies (62.0%) and prepectoral implant M-IBR in 324 (38.0%), with a significant increase in prepectoral placement according to the consecutive years: 3.1%, 5.5%, 40.1%, 63.5% and 61.7% in years 2019, 2020, 2021, 2022 and 2023, respectively (*p* < 0.0001).

Bilateral mastectomies were performed in 85 patients (170 mastectomies) with no significant difference between subpectoral and prepectoral implant M-IBR: 108 bilateral mastectomies for prophylactic purposes, 61 for primary BC and 1 for local recurrence (8 patients with mastectomy for primary BC on one side and prophylactic on the other side; 1 patient with mastectomy for primary BC on one side and local recurrence on the other side).

A mesh was concomitantly used in 9 subpectoral mastectomies (1.7%) and 176 prepectoral mastectomies (54.3%) with significantly different rates according to the year: 20.0%, 14.3%, 92.0%, 85.0% and 6.7% for prepectoral implants in years 2019, 2020, 2021, 2022 and 2023, respectively (*p* < 0.0001).

Patient characteristics according to subpectoral or prepectoral implant M-IBR are shown in Table 1. The type of mastectomy, previous ipsilateral surgery, neo-adjuvant chemotherapy, surgical incisions and surgeons involved were significantly different between the two groups. We could observe that within the same team of breast surgeons, the use of prepectoral implants widely varied, ranging from 0% to 60% (surgeon 1 placed 90 prepectoral implants out of 150 implantations). Median values and 95%CI are shown in Table 2, with no significant difference in age, BMI or mastectomy weight, but there is a significant difference in implant volume (higher volume in the prepectoral position) (*p* = 0.003).

Regression analysis showed significant prepectoral implant placement in the last 3 years, which was used less frequently in patients with previous breast surgery and by some surgeons. There was no significant difference between patients with or without NAC (Table 3).

### 3.1. Complications

One hundred and forty-seven mastectomies (17.2%) showed any grade of complication, with a significantly higher rate in prepectoral implant M-IBR (20.4%: 66/324) compared to the subpectoral position (15.3%: 81/529) (*p* = 0.036). However, there was no difference between the two groups for grade 2–3 complications (*p* = 0.097: 13.0% versus 9.8%), the implant loss rate (*p* = 0.271: 6.5% versus 4.7%) and the re-operation rate (*p* = 0.056: 10.8% versus 7.4%) (Table 1). The most common complications were poor blood supply or necrosis of the skin or nipple are-olar complex (45.6% of complications), hematoma (30.1%) and infection (16.2%), with no significant difference between subpectoral and prepectoral implant M-IBR (*p* = 0.179) (Table 1).

In terms of regression analysis, prepectoral implant was not significantly associated with any grade of complications (Table 4). Higher complications were observed in smokers (OR = 1.713, *p* = 0.022) and areolar and inverted T incisions (OR = 8.431, *p* = 0.004 and OR = 6.794, *p* = 0.004, respectively) and lower complications were observed in SSM (OR = 0.394, *p* = 0.035).

In addition, prepectoral implant was not significantly associated with grade 2–3 complications (Table 4). A higher rate of grade 2–3 complications was observed in smokers (OR = 1.844, *p* = 0.022), mesh use (OR = 2.194, *p* = 0.023), mastectomy weight > 300 gr (OR = 2.125, *p* = 0.002), diabetes (OR = 5.053, *p* = 0.046), mastectomy for local recurrence (OR = 2.645, *p* = 0.009) and concomitant sentinel lymph node biopsy (SLNB) (OR = 2.240, *p* = 0.016). There was no difference between patients with or without NAC.

### 3.2. Duration of Surgery

The median duration of surgery was 100 min, 105 mns for the subpectoral position (CI 95%: 104.8–110.8) and 90.5 mns for the prepectoral position (CI 95%: 93.0–99.8) (*p* < 0.0001). Regression analysis was evaluated for the duration of surgery > 120 min (260 patients: 30.5%) or ≤120 min (593 patients: 69.5%): prepectoral implant-M-IBR was associated with significantly shorter surgery duration (OR = 0.410, *p* = 0.002) and a significantly longer surgery duration was observed for mastectomy associated with SLNB and ALND, breast cup size > C and in three surgeons. Inferior breast fold incision and prophylactic mastectomy were associated with shorter operative times (Table 5).

### 3.3. Length of Postoperative Stay (LOS)

The median LOS was 1 day and was significantly lower with prepectoral implant placement (Table 2). Regression analysis was evaluated for LOS < or >2 days. Prepectoral implant placement was not significantly associated with longer LOS. A significant association with longer LOS was observed for prophylactic mastectomy and mastectomy weight > 300 gr. Shorter LOS was observed in the years 2020 to 2023 compared to the year 2019 (Table 6).

### 3.4. Adjuvant Therapy

NAC was administered in 13.5% of patients (115/853) with a significantly higher rate in the prepectoral implant M-IBR group (Table 1). Adjuvant chemotherapy was administered in 19.86% of patients (139/700), with a significantly higher rate in the subpectoral implant M-IBR group (Table 1). Twenty-five percent of patients received PMRT, with no significant difference between subpectoral and prepectoral implant-M-IBR (Table 1). Sixty-two percent of primary BC patients received endocrine therapy, with no significant difference between subpectoral and prepectoral implant-M-IBR (Table 1).

The median time interval between surgery and adjuvant treatment was 48 days (mean: 54.6; CI 95%: 50.4–58.7; range: 11–291): 47 days (mean: 54.7; CI 95%: 48.9–60.6; range: 15–291) in the subpectoral implant group (129 patients) and 49 days (mean: 54.6; CI 95%: 49.1–60.1; range: 11–131) in the prepectoral implant group (75 patients) (*p* = 0.993).

The occurrence of complications did not affect the median time interval to adjuvant therapy: it was 48 days (mean: 55.0; CI 95%: 50.3–59.7; range: 11–291), 47 days (mean: 52.3; CI 95%: 43.6–61.0; range: 11–130), 48 days (mean: 54.0; CI 95%: 49.5–58.5; range: 11–291) and 55.5 days (mean: 59.2; CI 95%: 47.3–71.1; range: 27–130) in patients without no complication (171 patients), one complication of any grade (34 patients), no complication of grade 2–3 (183 patients) and one complication of grade 2–3 (22 patients), respectively (*p* = 0.451).

The median time interval between surgery and adjuvant chemotherapy (125 patients) was 43 days (mean: 45.1; CI 95%: 42.1–48.1; range: 11–122) and was 60 days (mean: 69.9; CI 95%: 61.0–78.7; range: 29–291) between surgery and PMRT (79 patients) (*p* < 0.0001).

### 3.5. Satisfaction

Good or very good satisfaction was observed in 74.2% (633/853) and failure-bad-medium satisfaction in 25.8% (220/853) (Table 1), with significant differences according to several factors: complications of all grades (*p* < 0.0001), grade 2–3 complications (*p* < 0.0001), indication (*p* < 0.0001), years (*p* < 0.0001), mastectomy weight (*p* = 0.030), axillary surgery (*p* = 0.016), previous radiotherapy (*p* < 0.0001) and smoking status (*p* = 0.029).

Patient satisfaction appeared to be independent of the surgeon (*p* = 0.070), the type of mastectomy (*p* = 0.093), the type of implant (*p* = 0.059), ASA status (*p* = 0.054), the implant size (*p* = 0.387), BMI (*p* = 0.693), incisions (*p* = 0.115), the breast cup size (*p* = 0.216), NAC (*p* = 0.333), previous ipsilateral surgery (*p* = 0.353), the implant position (*p* = 0.207), the mesh (*p* = 0.296) and age (*p* = 0.057).

In regression analysis, factors significantly associated with lower satisfaction (failure-bad-medium) were the year 2020 and grade 2–3 complications (Table 7).

### 3.6. Cost Evaluation

The median cost was 4178 Euros (mean: 4342; CI 95%: 4240–4444; range: 2788–14849): 3560 Euros (mean: 4236; CI 95%: 4101–4371; range: 2828–14849) for a subpectoral implant and 4426 Euros (mean: 4515; CI 95%: 4360–4670; range: 2788–12166) for a prepectoral implant.

For the prepectoral implant position, the median cost was 3305 Euros (mean: 3876; CI 95%: 3668–4085; range: 2788–9032) without a mesh and 4567 Euros (mean: 5053; CI 95%: 4859–5246; range: 4178–12166) with a mesh.

While the cost of a pre-pectoral implant procedure was 255 euros less expensive than a subpectoral implant procedure (−7.7%), the addition of mesh to the pre-pectoral implant procedure increased the cost by 1262 euros (38.2%).

In regression analysis, factors significantly associated with a cost higher than the median cost (4178 Euros) were: subpectoral placement (OR: 1.603, CI95% 1.070–2.400, *p* = 0.022) and mesh use (OR: 234.7, CI95% 55.7–989.9, *p* < 0.0001), while SLNB, on the other hand, generated a lower cost than no axillary surgery (OR: 0.557, CI95% 0.399–0.777, *p* = 0.001) (no significant difference between ALND and no axillary surgery: OR: 0.701, CI95% 0.378–1.301, *p* = 0.2660). Breast cup size had no significant effect on costs. Taking into account the shorter LOS for the prepectoral implant, it resulted in lower costs than the subpectoral implant and higher costs than the prepectoral implant IBR with a mesh.

### 3.7. Scores

The calculation of predictive scores for any complication or grade 2–3 complication using the ORs of significant factors from the regression analysis isolated four and five risk groups, respectively. A higher score predicted a higher risk of events, but with a low AUC value (<0.70).

### 3.8. Satisfaction and Complications According to Score Groups

In patients scored by the “any complication score”, good-very good satisfaction was reported by 77.7% (209/269), 75.2% (261/347), 67.9% (72/106) and 69.5% (91/131) in groups 1 to 4, respectively (*p* = 0.132). In group 3, 55.6% (20/36) of patients with a subpectoral implant were satisfied, compared with 74.3% (52/70) of those with a prepectoral implant (*p* = 0.042) (Table 8).

In patients scored by the “grade 2–3 complication score”, good-very good satisfaction was reported by 80.0% (348/435), 71.7% (198/276), 67.9% (74/109), 30.4% (7/23) and 60.0% (6/10) in groups 1 to 5, respectively (*p* < 0.0001). In group 2, 67.5% (108/160) of patients with a subpectoral implant were satisfied compared to 77.6% (90/116) of those with a prepectoral implant (*p* = 0.044) (Table 8).

In patients with an “any grade complication” score rating of 4, those with a prepectoral implant had significantly fewer complications than those with a subpectoral implant. Prepectoral implant mesh patients with an “any grade complication” score rating of 3 had a significantly lower risk of experiencing complications than patients with a score of 3, patients with a prepectoral implant without a mesh. In patients scored with the “grade 2–3 complications” score, no significant difference was observed (Table 8).

## 4. Discussion

M-IBR has increased in recent years, particularly implant IBR [18] and, more recently, prepectoral implant M-IBR has shown a rapid adoption, as reported by Chinta et al. [45]. We reported a significant increase in prepectoral M-IBR since the year 2021 but with significant variation in practice between surgeons. The use of mesh also increased in 2021–2022 (92% and 85% for prepectoral implant-M-IBR), but sharply decreased in 2023 (6.7%).

### 4.1. Complications

We report a high rate of skin and NAC complications, no doubt related to the high rate of NSM (52.9%). There is a balance between the risks of breast recurrence, complications, and final cosmetic results. The more fat tissue that remains, the lower the complication rate and the better the cosmetic result will be, but conversely, the more breast tissue that remains, the higher the risk of local recurrence [46]. Sixty to eighty percent of local recurrences have been reported to be located within the skin, the nipple-areolar complex (NACx) and subcutaneous tissue [47,48]. In MRI studies, the rate of residual glandular tissue reached 20%, higher in NSM than in SSM [49]. The thickness of the skin flap is of paramount importance. Andersson et al. [50] reported residual glandular tissue after prophylactic mastectomy in 6.9% of skin flaps ≤ 5 mm and 37.5% of skin flaps > 5 mm (OR 3.07; *p* = 0.005) with a significant increase when the flap thickness exceeded 7 mm (more than 40%). Consequently, complete breast tissue removal is required but at the cost of an increased risk of ischemic mastectomy flaps: a flap thickness of less than 5 or 8 mm has been reported as an independent predictor of ischemic complications [51,52] with odds of skin necrosis six times higher in skin flaps ≤ 5 mm compared to >5 mm [53]. Locoregional recurrence after NSM has been reported to be 0–11.7% and recurrence within the NAC itself has been reported at 0–5% [54]. In a recent systematic review [55], including 19 studies with 1917 implant M-IBR, local recurrence rates were localized in 4.7% of the cases in the skin, 0.4% in the chest wall and 0.4% in the NACx. The context of the mastectomy is also important. The local recurrence rate was found at 7.9% to 11.4% after therapeutic mastectomy [56,57] and 0 to 1.6% after prophylactic mastectomy [46,47,58,59,60,61,62]. The reported local recurrence rate within the NACx is very low and the rate of NACx necrosis is usually less than 11% [19,24,63,64,65]. The timing of breast reconstruction (immediate or delayed) [66,67], breast reconstruction per se [68] and implant location [69] did not affect local recurrence rates.

We observed an “any grade” complication rate of 17.2% which favorably compares with the literature data in which complication rates range from 19% to 42% depending on the implant-based reconstruction method (mesh or no mesh, NSM or SSM) [70,71,72,73,74]. In univariate analysis, the “any grade” complication rate was higher with prepectoral implants but was no longer significant in multivariate analysis. As we reported, prepectoral implants were associated with the use of a mesh in 54.3% of the cases (in contrast to subpectoral implants where a mesh was used in only 1.7% of patients). It has been reported that the use of a mesh has a significant and negative impact on the complication rate. As a result, the use of a mesh has dramatically fallen. In fact, a recent meta-analysis of 17 studies, evaluated complication rates comparing the acellular dermal matrix (ADM), a synthetic absorbable mesh, a synthetic non-absorbable mesh and no matrix [75]. The infection rate was higher for ADM, the seroma rate was lower for the synthetic absorbable matrix (OR: 0.2) and the synthetic non-absorbable matrix (OR: 0.1 compared to ADM). However, clinically significant complications (grade 2 and 3) did not differ between patients who received prepectoral or subpectoral implants [75]. In the above meta-analysis, major complications were similar whether or not a mesh was used. In another recent meta-analysis including 31 studies, there was no significant difference in overall complications and implant loss between subpectoral implantation without a mesh and a xenograft acellular dermal matrix (OR: 0.63) and synthetic mesh (OR: 0.77) [76]. On the other hand, areolar incisions and inverted T-incisions were significantly associated with a higher risk of overall complications, mirroring a report by Frey et al. [77] with a lower risk of mastectomy flap necrosis for inframammary fold incisions. Due to the small number of PMRT, we were not able to compare the effect in prepectoral versus subpectoral implants, but early literature data seems to favor the prepectoral position with less capsular contracture, probably due to a lower inflammation burden [78,79,80,81]. In addition, other studies have reported that neither neoadjuvant nor adjuvant chemotherapy was associated with the likelihood of complications in patients undergoing implant reconstruction, regardless of the implant position [82,83]. Finally, we defined different complication risk scores to compare mastectomy and reconstruction techniques in the hope of facilitating comparisons between centers and studies. The key issue in clinical practice is the occurrence of grade 2–3 complications, which significantly increase with the risk group, but with a low predictive value (0.678).

To summarize, implant positioning (prepectoral or subpectoral) has no effect on the clinically relevant complication rate; only mesh use, smoking status and incision type (areolar and inverted T) had a negative effect. Moreover, the thickness and quality of perfusion of the skin flap are major factors in skin necrosis and implant loss and are the main criteria for deciding on prepectoral implantation [84,85]. Careful patient selection is therefore an important factor in minimizing complications. Obesity, smoking and diabetes are all known to be associated with complications [86,87,88] and the iBRA study confirmed the link between infection, previous radiation, prolonged operative time and the need for reoperation [39]. Complication risk scores may help to select patients for whom implant M-IBR is proposed.

### 4.2. Time to Adjuvant Therapies

Interestingly, the occurrence of complications, even grade 2–3 complications, did not significantly affect the time to start adjuvant treatment. The median time interval between surgery and adjuvant chemotherapy was 43 days and 60 days for PMRT. In a meta-analysis, Cook et al. reported an increase mean time to chemotherapy of 3.50 days, from 40.38 days after surgery to 43.56 days for mastectomy without IBR and with IBR, respectively, without clinical significance [89]. O’Connell et al. reported that major complications significantly increased treatment delays [90], with no significant effect on BC recurrence and death rates [91].

### 4.3. Patients’ Satisfaction, Length of Surgery and Costs

Patient satisfaction was independent of the implant location but strongly associated with the occurrence of complications (*p* < 0.0001). The use of mesh did not change this measure.

As reported by others [45], prepectoral implants shortened the duration of surgery (OR = 0.410, *p* = 0.002). Although the operative time was shorter for prepectoral implantation, Chinta et al. found no significant difference in cost between prepectoral and subpectoral implants in a multivariate analysis. In the regression analysis, we reported higher costs for subpectoral implantation (OR: 1.603) and for mesh use (OR: 234.7). Prepectoral reconstruction was associated with higher operative costs, undoubtedly due to the additional cost of the ADM [84]. However, the use of mesh has gradually decreased over the years in our practice and prepectoral M-IBR without mesh use cost 25% less than subpectoral placement in a comparative study [92]. Moreover, cost evaluation with percentage differences between different techniques seems to be more accurate than the quantitative value, as the coverage of the cost of interventions varies between countries and the economic models.

### 4.4. Limitations

Our study has several limitations. First, the retrospective nature of the study with potential biases despite multivariate analysis and a single-center study the results of which may not be generalizable to other teams. Second, only the short-term outcomes were examined. Thirdly, the relevance of the cost analysis to the particular center, as the cost calculation would vary from center to center/country to country. Finally, satisfaction was assessed without pre- and post-operative quality of life evaluation with the same follow-up. Future studies including several centers are needed and we plan a multicenter study with a larger number of patients, different practices and the use of different mesh types.

## 5. Conclusions

The results of our study provide further evidence that prepectoral implants do not lead to higher complication rates, are shorter and less expensive, provided that no mesh is used. Furthermore, compared with previous studies, we showed that the time to adjuvant therapy does not differ from subpectoral implants and patient satisfaction remains similar. Prepectoral implantation can be considered a good and safe technique. A complication risk score may help in the decision to implant-M-IBR and may help to compare results between different teams and studies. These results need to be confirmed in other centers, and the predictive score for complications needs to be improved, in particular through a larger multicenter study that is currently underway.

## Figures and Tables

**Table 1 cancers-16-01129-t001:** Characteristics of patients according to sub-pectoral or pre-pectoral implant M-IBR.

		Sub Pectoral		Pre Pectoral		Chi2	Total	
		Nb	%	Nb	%	*p*	Nb	%
Indication	Primary BC	387	73.2	246	75.9	0.652	633	74.2
	Local recurrence	44	8.3	23	7.1		67	7.9
	Prophylactic	98	18.5	55	17.0		153	17.9
Mesh	No	520	98.3	148	45.7	<0.0001	668	78.3
	Yes	9	1.7	176	54.3		185	21.7
Mastectomy	NSM	236	44.6	215	66.4	<0.0001	451	52.9
	SSM	290	54.8	107	33.0		397	46.5
	standard	3	0.6	2	0.6		5	0.6
Type reconstruction	definitive implant	501	94.7	324	100	<0.0001	824	96.6
	expander	28	5.3	0	0		28	3.3
bilateral mastectomy	No	428	80.9	255	78.7	0.243	683	78.7
	Yes	101	19.1	69	21.3		170	21.3
ASA status	1	247	46.7	152	46.9	0.728	399	46.8
	2	273	51.6	164	50.6		437	51.2
	3	9	1.7	8	2.5		17	2.0
Breast cup size	A-B	278	52.6	177	54.6	0.782	455	53.3
	C	173	32.7	104	32.1		277	32.5
	>C	78	14.7	43	13.3		121	14.2
Tobacco	No	437	82.6	268	82.7	0.968	705	82.6
	Yes	92	17.4	56	17.3		148	17.4
Diabetes	No	525	99.2	321	99.1	0.536 *	846	99.2
	Yes	4	0.8	3	0.9		7	0.8
Previous surgery	No	331	62.6	238	73.5	0.001	569	66.7
	Yes	198	37.4	86	26.5		284	33.3
NAC	No	469	88.7	269	83.0	0.013	738	86.5
	Yes	60	11.3	55	17.0		115	13.5
Previous radiotherapy	No	476	90.0	292	90.1	0.523	768	90.0
	Yes	53	10.0	32	9.9		85	10.0
Complication	No	448	84.7	258	79.6	0.036 *	706	82.8
	Yes	81	15.3	66	20.4		147	17.2
G2–3 complication	No	477	90.2	282	87.0	0.097	759	89.0
	Yes	52	9.8	42	13.0		84	11.0
Implant loss	No	504	95.3	303	93.5	0.271	807	94.6
	Yes	25	4.7	21	6.5		46	5.4
Re-operation	No	490	92.6	289	89.2	0.056	779	91.3
	Yes	39	7.4	35	10.8		74	8.7
Surgeon	1	60	11.3	90	27.8	<0.0001	150	17.6
	2	82	15.5	15	4.6		97	11.4
	3	117	22.1	40	12.3		157	18.4
	4	50	9.5	0	0		50	5.9
	5	74	14.0	50	15.4		124	14.5
	6	40	7.6	48	14.8		88	10.3
	7	69	13.0	1	0.3		70	8.2
	8	11	2.1	6	1.9		17	2.0
	9	15	2.8	18	5.6		33	3.9
	10	7	1.3	56	17.3		63	7.4
	11	4	0.8	0	0		4	0.5
Complication type	skin—NACx	37	48.7	25	41.7	0.179	62	45.6
	hematoma	21	27.6	20	33.3		41	30.1
	infection	13	17.1	9	15.0		22	16.2
	lymphocel	2	2.6	6	75.0		8	5.9
	others	3	3.9	0	0		3	2.2
Satisfaction	failure	29	5.5	22	6.8	0.346	51	6.0
	bad	17	3.2	13	4.0		30	3.5
	medium	96	18.1	43	13.3		139	16.3
	good	263	49.7	173	53.4		436	51.1
	very good	124	23.4	73	22.5		197	23.1
Interval time to	≤60 days	96	73.8	52	69.3	0.487	148	72.2
adjuvant therapy	>60 days	34	26.2	23	30.7		57	27.8
Surgical incision	axillary	29	5.5	12	3.7	<0.0001	41	4.8
	areolar	16	3.0	2	0.6		18	2.1
	central	269	50.9	107	33.0		376	44.1
	previous incision	12	2.3	4	1.2		16	1.9
	inversed T	23	4.3	4	1.2		27	3.2
	areolar + radial	83	15.7	54	16.7		137	16.1
	radial	11	2.1	3	0.9		14	1.6
	inferior fold	86	16.3	138	42.6		224	26.3
LOS	1 day	295	55.8	239	73.8	<0.0001	534	62.6
	2	146	27.6	59	18.2		205	24.0
	3	66	12.5	13	4.0		79	9.3
	4	11	2.1	6	1.9		17	2.0
	5	7	1.3	5	1.5		12	1.4
	>5	4	0.8	2	0.6		6	0.6
Axillary surgery	No	211	39.9	129	39.8	0.156	340	39.9
	SLNB	279	52.7	159	49.1		438	51.3
	ALND	39	7.4	36	11.1		75	8.8

Legend: M-IBR: mastectomy immediate breast reconstruction, BC: breast cancer, NSM: Nipple-sparing mastectomy, SSM: Skin-sparing mastectomy, ASA: American Society of Anesthesiologists, NAC: neo-adjuvant chemotherapy, NACx: nipple areolar complex, LOS: Length of post-operative stay, SLNB: sentinel lymph node biopsy, ALND: axillary lymph node dissection. *: Fisher test.

**Table 2 cancers-16-01129-t002:** Characteristics according to sub-pectoral or pre-pectoral implant-M-IBR: Median values and CI 95%.

	Sub Pectoral		Pre Pectoral		*t*-Test	Total	
	Value	CI 95%	Value	CI 95%	*p*	Value	CI 95%
Median age	48.0	48.46–50.50	48.0	48.82–51.57	0.408	48.0	48.94–50.57
Median BMI	22.26	22.76–23.41	22.74	22.73–23.51	0.882	22.49	22.85–23.34
Median PLOH	1.0	1.60–1.78	1.0	1.32–1.50	<0.0001	1.0	1.52–1.65
Median Length Surgery	105	104.8–110.8	90.5	93.0–99.8	<0.0001	100	101.2–105.8
Median mastectomy weight	308	328–363	315	338–383	0.304	310	337–365
Median implant volume	300	289–308	300	309–332	0.003	300	300–314
Median cost	3560	4101–4371	4426	4360–4670	0.009	4178	4240–4444
Median cost without mesh	3513	4037–4265	3305	3668–4084	0.025	3442	3990–4190
Median cost with mesh	7752	5463–12848	4567	4859–5246	<0.0001	4581	4985–5519

Legend: PLOH: Length of post-operative hospitalization.

**Table 3 cancers-16-01129-t003:** Factors associated with pre-pectoral versus sub-pectoral implant IBR: regression analysis.

Pre vs. Sub-Pectoral: Regression			*p*	OR	CI 95%	
		Nb			Inferior	Superior
Mastectomy	NSM	451	0.076	1		
	SSM	397	0.023	0.603	0.390	0.933
	Standard	5	0.871	0.783	0.040	15.152
Implant type	Expander vs. definitive	28/825	0.998	<0.0001	0.000	
Years	2019	167	<0.0001	1		
	2020	128	0.325	1.708	0.589	4.955
	2021	187	<0.0001	17.292	7.806	38.305
	2022	178	<0.0001	148.59	57.052	387.03
	2023	193	<0.0001	209.79	73.340	600.13
Surgeon	1	150	<0.0001	1		
	2	97	<0.0001	0.017	0.006	0.047
	3	157	<0.0001	0.187	0.089	0.393
	4	50	<0.0001	0.000	0.000	
	5	124	<0.0001	0.457	0.230	0.909
	6	88	<0.0001	0.163	0.067	0.397
	7	70	<0.0001	0.001	0.000	0.009
	8	17	0.002	0.085	0.018	0.405
	9	33	<0.0001	0.037	0.012	0.118
	10	63	0.025	0.275	0.089	0.850
	11	4	0.999	0.000	0.000	
Previous surgery	Yes vs. No	284/569	0.005	0.517	0.326	0.821
NAC	Yes vs. No	115/738	0.742	1.103	0.615	1.979

Legend: M-IBR: Mastectomy immediate breast reconstruction, NSM: Nipple-sparing mastectomy, SSM: Skin-sparing mastectomy, NAC: neo-adjuvant chemotherapy.

**Table 4 cancers-16-01129-t004:** Factors associated with all complications and grade 2–3 complications according to pre-pectoral versus sub-pectoral implant placement in regression analysis.

All Complications: Regression			*p*	OR	CI 95%	
		Nb			Inferior	Superior
Incision	axillar	41	<0.0001	1		
	areolar	18	0.004	8.431	2.011	35.336
	central	376	0.300	1.978	0.545	7.172
	previous	16	0.992	0.991	0.159	6.173
	inversed T	27	0.004	6.794	1.816	25.423
	areolar + radial	137	0.060	2.717	0.958	7.709
	radial	14	0.563	0.511	0.053	4.961
	inferior fold	224	0.698	1.233	0.428	3.553
Breast Cup size	A-B	455	0.346	1		
	C	277	0.366	1.243	0.776	1.993
	>C	121	0.152	1.538	0.853	2.771
Smokers	Yes vs. No	148/705	0.022	1.713	1.079	2.718
ASA status	1	399	0.178	1		
	2	437	0.193	1.295	0.877	1.910
	3	17	0.118	2.526	0.791	8.064
Mesh	Yes vs. No	185/668	0.002	2.558	1.392	4.702
Mastectomy	NSM	451	0.065	1		
	SSM	397	0.035	0.394	0.166	0.935
	Standard	5	0.808	1.300	0.158	10.716
Implant position	Pre vs. Sub	324/529	0.529	0.846	0.502	1.424
Mastectomy weight	>vs. ≤ 300	448/405	0.111	1.451	0.918	2.292
**Grade 2–3 complication: regression**			** *p* **	**OR**	**CI 95%**	
		**Nb**			**Inferior**	**Superior**
Smoker	Yes vs. No	148/705	0.022	1.844	1.094	3.107
Mesh	Yes vs. No	185/668	0.023	2.194	1.117	4.309
Implant position	Pre vs. Sub	324/529	0.784	0.916	0.491	1.711
Mastectomy weight	> vs. ≤300 gr	448/405	0.002	2.125	1.320	3.422
NAC	Yes vs. No	115/738	0.113	0.477	0.190	1.193
Diabetes	Yes vs. No	7/846	0.046	5.053	1.030	24.789
Indication	Primary BC	633	0.031	1		
	Recurrence	67	0.009	2.645	1.281	5.461
	Prophylactic	153	0.444	1.397	0.594	3.284
Axillary surgery	No	340	0.048	1		
	SLNB	438	0.016	2.240	1.160	4.328
	ALND	75	0.491	1.419	0.524	3.841

Legend: BC: breast cancer, NSM: Nipple-sparing mastectomy, SSM: Skin-sparing mastectomy, ASA: American Society of Anesthesiologists, NAC: neo-adjuvant chemotherapy, SLNB: sentinel lymph node biopsy, ALND: axillary lymph node dissection.

**Table 5 cancers-16-01129-t005:** Regression analysis for length of surgery >120 min or ≤120 min.

Length of Surgery: Regression			*p*	OR	CI 95%	
		Nb			Inferior	Superior
Mesh	Yes vs. No	185/668	0.655	1.192	0.551	2.579
Implant position	Pre vs. Sub	324/529	0.011	0.426	0.221	0.821
Mastectomy weight	> vs. ≤300 gr	448/405	0.559	1.149	0.721	1.832
Indication	Primary BC	633	0.127	1		
	Recurrence	67	0.942	1.028	0.494	2.136
	Prophylactic	153	0.046	0.467	0.221	0.986
Axillary surgery	No	340	<0.0001	1		
	SLNB	438	0.001	2.458	1.479	4.087
	ALND	75	<0.0001	8.911	4.299	18.470
Breast Cup Size	A-B	455	0.116	1		
	C	277	0.337	1.254	0.791	1.987
	>C	121	0.038	1.948	1.038	3.658
BMI	≤24.9	626	0.143	1		
	25–29.99	188	0.602	1.136	0.704	1.831
	≥30	39	0.049	2.486	1.006	6.145
Surgeon	1	150	<0.0001	1		
	2	97	<0.0001	4.645	2.108	10.233
	3	157	0.109	0.521	0.235	1.157
	4	50	0.156	2.015	0.766	5.299
	5	124	0.722	1.153	0.527	2.524
	6	88	0.820	1.102	0.476	2.550
	7	70	<0.0001	5.746	2.368	13.940
	8	17	0.005	6.280	1.758	22.434
	9	33	0.157	2.222	0.735	6.714
	10	63	<0.0001	6.109	2.522	14.799
	11	4	0.912	0.866	0.068	11.067
Years	2019	167	0.809	1		
	2020	128	0.953	0.980	0.501	1.916
	2021	187	0.900	0.958	0.494	1.860
	2022	178	0.483	1.305	0.621	2.740
	2023	193	0.726	0.873	0.407	1.872
Mastectomy	NSM	451	0.996	1		
	SSM	397	0.931	0.962	0.400	2.313
	Standard	5	0.994	0.989	0.062	15.738
incision	axillar	41	<0.0001	1		
	areolar	18	<0.0001	0.022	0.004	0.122
	central	376	<0.0001	0.010	0.002	0.047
	previous	16	<0.0001	0.011	0.002	0.069
	inversed T	27	<0.0001	0.014	0.0022	0.079
	areolar + radial	137	<0.0001	0.012	0.003	0.049
	radial	14	<0.0001	0.013	0.002	0.094
	inferior fold	224	<0.0001	0.008	0.002	0.031

Legend: BC: breast cancer, NSM: Nipple-sparing mastectomy, SSM: Skin-sparing mastectomy, SLNB: sentinel lymph node biopsy, ALND: axillary lymph node dissection, BMI: body mass index.

**Table 6 cancers-16-01129-t006:** Regression analysis for length of post-operative stay (LOS) ≤ or >2 days.

LOS ≤ versus >2 Days: Regression			*p*	OR	CI 95%	
		Nb			Inferior	Superior
BMI	≤24.9	626	0.083	1		
	25–29.99	188	0.117	0.613	0.332	1.130
	≥30	39	0.251	1.642	0.704	3.829
Mesh	Yes vs. No	185/668	0.599	0.786	0.320	1.930
Indication	Primary BC	633	0.004	1		
	Recurrence	67	0.671	1.197	0.521	2.750
	Prophylactic	153	0.001	2.557	1.463	4.469
Years	2019	167	<0.0001	1		
	2020	128	<0.0001	0.213	0.099	0.455
	2021	187	<0.0001	0.264	0.129	0.543
	2022	178	0.001	0.276	0.125	0.607
	2023	193	0.014	0.382	0.178	0.820
Mastectomy	NSM	451	0.559	1		
	SSM	397	0.494	0.705	0.258	1.922
	Standard	5	0.543	2.125	0.187	24.163
Implant position	Pre vs. Sub	324/529	0.202	0.629	0.309	1.282
incision	inferior fold	224	0.379	1		
	axillar	41	0.466	1.395	0.570	3.415
	areolar	18	0.225	2.158	0.623	7.472
	central	376	0.683	0.794	0.262	2.408
	previous	16	0.697	1.339	0.308	5.823
	inversed T	27	0.246	1.936	0.633	5.917
	areolar + radial	137	0.358	0.705	0.335	1.485
	radial	14	0.883	0.882	0.166	4.693
Mastectomy weight	> vs. ≤300 gr	448/405	0.001	2.315	1.398	3.833

Legend: BC: breast cancer, NSM: Nipple-sparing mastectomy, SSM: Skin-sparing mastectomy, BMI: body mass index.

**Table 7 cancers-16-01129-t007:** Factors associated with less satisfaction (failure-bad-medium) versus good and very good satisfaction: regression analysis.

Satisfaction: Regression			*p*	OR	CI 95%	
		Nb			Inferior	Superior
Indication	Primary BC	633	0.512	1		
	Recurrence	67	0.615	1.300	0.468	3.608
	Prophylactic	153	0.361	0.755	0.413	1.380
Years	2019	167	<0.0001	1		
	2020	128	0.017	1.916	1.121	3.275
	2021	187	0.277	1.319	0.801	2.174
	2022	178	0.041	0.550	0.310	0.975
	2023	193	0.325	0.767	0.453	1.300
Mastectomy weight	> vs. ≤300 gr	448/405	0.425	1.150	0.816	1.622
Axillary surgery	No	340	0.189	1		
	SLNB	438	0.068	1.506	0.970	2.338
	ALND	75	0.365	1.368	0.694	2.694
Previous radiotherapy	Yes vs. No	85/768	0.108	2.113	0.849	5.258
Smoker	Yes vs. No	148/705	0.292	1.260	0.820	1.937
G 2–3 complication	Yes vs. No	94/759	<0.0001	6.230	3.858	10.060

Legend: BC: breast cancer, SLNB: sentinel lymph node biopsy, ALND: axillary lymph node dissection.

**Table 8 cancers-16-01129-t008:** Satisfaction and complications according to pre-pectoral or sub-pectoral implant placement and with or without a mesh in each group at risk of complications.

			Satisfaction		Chi2	Complications		Chi2
			Nb	%	*p*	Nb	%	*p*
**Score all complications**			**Good-very good**			**All complications**		
Score 1	Sub-pectoral		175/230	76.1	0.087	(27/230)	11.7	0.510 *
	Pre-pectoral		34/39	87.2		(5/39)	12.8	
	Pre-pectoral	without mesh	34/39	87.2		(5/39)	12.8	
	Pre-pectoral	with mesh	0	0		0	0	
Score 2	Sub-pectoral		189/256	73.8	0.195	(38/256)	14.8	0.525 *
	Pre-pectoral		72/91	79.1		(13/91)	14.3	
	Pre-pectoral	without mesh	72/91	79.1		(13/91)	14.3	
	Pre-pectoral	with mesh	0	0		0	0	
Score 3	Sub-pectoral		20/36	55.6	0.042	(11/36)	30.6	0.126 *
	Pre-pectoral		52/70	74.3		(13/70)	18.6	
	Pre-pectoral	without mesh	(10/18)	55.6	0.039	(7/18)	38.9	0.016
	Pre-pectoral	with mesh	42/52	80.8		(6/52)	11.5	
Score 4	Sub-pectoral		(3/7)	42.9	0.127	(5/7)	71.4	0.027 *
	Pre-pectoral		88/124	71.0		(35/124)	28.2	
	Pre-pectoral	without mesh	88/124	71.0		(35/124)	28.2	
	Pre-pectoral	with mesh	0	0		0	0	
**Score Grade 2–3 complications**			**Good-very good**			**Grade 2–3 complications**		
Score 1	Sub-pectoral		249/315	79.0	0.254	(21/315)	6.7	0.229 *
	Pre-pectoral		99/120	82.5		(5/120)	4.2	
	Pre-pectoral	without mesh	76/94	80.9	0.279 *	(5/94)	5.3	0.288 *
	Pre-pectoral	with mesh	23/26	88.5		(0/26)	0	
Score 2	Sub-pectoral		108/160	67.5	0.044	(17/160)	10.6	0.425
	Pre-pectoral		90/116	77.6		(14/116)	12.1	
	Pre-pectoral	without mesh	30/37	81.1	0.358 *	(5/37)	13.5	0.480 *
	Pre-pectoral	with mesh	60/79	75.9		(9/79)	11.4	
Score 3	Sub-pectoral		25/40	62.5	0.240	(7/40)	17.5	0.393
	Pre-pectoral		49/69	71.0		(15/69)	21.7	
	Pre-pectoral	without mesh	(8/14)	57.1	0.170 *	(4/14)	28.6	0.357 *
	Pre-pectoral	with mesh	41/55	74.5		(11/55)	20.0	
Score 4	Sub-pectoral		(1/8)	12.5	0.190	(4/8)	50.0	0.611 *
	Pre-pectoral		(6/15)	40.0		(7/15)	46.7	
	Pre-pectoral	without mesh	(2/2)	100	0.143 *	(0/2)	0	0.267 *
	Pre-pectoral	with mesh	(4/13)	30.8		(7/13)	53.8	
Score 5	Sub-pectoral		(4/6)	66.7	0.548	(3/6)	50.0	0.452 *
	Pre-pectoral		(2/4)	50.0		(1/4)	25.0	
	Pre-pectoral	without mesh	(0/1)	0	0.500 *	(1/1)	100	0.250 *
	Pre-pectoral	with mesh	(2/3)	66.7		(0/3)	0	

Legend: *: Fisher test.

## Data Availability

Data supporting reported results can be found in Paoli Calmettes Institute breast cancer data base.

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
