# Peer review of "Postoperative Outcomes of Pre-Pectoral Versus Sub-Pectoral Implant Immediate Breast Reconstruction"

_cancers, 2024, doi:10.3390/cancers16061129_

Round 1
Reviewer 1 Report
Comments and Suggestions for Authors
The study performed by Houvenaeghel et al. presents a comprehensive analysis of implant-based mastectomy immediate breast reconstruction (M-IBR), focusing on post-operative complications, patient satisfaction, and costs associated with subpectoral versus prepectoral implant placements. It utilizes a large mono-centric cohort to establish a predictive score for post-operative complications. Key findings indicate no significant difference in clinically meaningful complication rates between pre- or subpectoral implant placements, with mesh use, smoking status, and incision type being major determinants of negative outcomes. Interestingly, the occurrence of complications did not significantly delay the initiation of adjuvant therapies.
I think that the study is novel and interesting; although, I think it needs incorporation of another retrospective analysis on neoadjuvant chemotherapy's impact on breast reconstruction outcomes (PMID: 36143318), to enhance the power of the discussion section.
Additionally, I suggest in the Limitations paragraph (page 15, lines 422-425) to remove that reference. Just present the limitations of your study in that paragraph.
The English language requires extensive editing for better understanding and clarification for the journal readers (especially in the simple summary and abstract).
Comments on the Quality of English LanguageThe English language requires extensive editing for better understanding and clarification for the journal readers (especially in the simple summary and abstract).
Author Response
Answer: thank you for this review and your comments
I think that the study is novel and interesting; although, I think it needs incorporation of another retrospective analysis on neoadjuvant chemotherapy's impact on breast reconstruction outcomes (PMID: 36143318), to enhance the power of the discussion section.
Answer: We have added line 165: There was no significant difference between patients with and without NAC. (Table 3) and in regression analysis for grade 2-3 complications, line 199: No difference was observed in patients with or without NAC.
In chapter discussion line 391-384, we have added 2 references and text: In addition, other studies have reported that neither neoadjuvant nor adjuvant chemotherapy was associated with the likelihood of complications in patients who underwent implant reconstruction, regardless of implant position [82-83].
Refrences:
Hart SE, Brown DL, Kim HM, Qi J, Hamill JB, Wilkins EG. Association of Clinical Complications of Chemotherapy and Patient-Reported Outcomes After Immediate Breast Reconstruction. JAMA Surg. 2021. 1;156(9):847-855. doi: 10.1001/jamasurg.
Scardina L, Di Leone A, Biondi E, Carnassale B, Sanchez AM, D'Archi S, Franco A, Moschella F, Magno S, Terribile D, Gentile D, Fabi A, D'Angelo A, Barone Adesi L, Visconti G, Salgarello M, Masetti R, Franceschini G. Prepectoral vs. Submuscular Immediate Breast Reconstruction in Patients Undergoing Mastectomy after Neoadjuvant Chemotherapy: Our Early Experience. J Pers Med. 2022 Sep 19;12(9):1533. doi: 10.3390/jpm12091533.
Additionally, I suggest in the Limitations paragraph (page 15, lines 422-425) to remove that reference. Just present the limitations of your study in that paragraph.
Answer: the phrase and reference "However, Sobti et al [91] reported that prepectoral implant-IBR is safe with lesser capsular contracture in irradiated patients than subpectoral implant placement." in chapter limitations has been deleted.
The English language requires extensive editing for better understanding and clarification for the journal readers (especially in the simple summary and abstract).
Answer: Simple summary and abstract have been corrected and English correction has been made.
Reviewer 2 Report
Comments and Suggestions for Authors
This study aimed to report the results of a single centre cohort of implant-IBR comparing subpectoral and prepectoral implant with or without mesh.
however there are some issues with this paper
1) pls explain the abbreviation of LPOH in line 5
2) for discussion- pls remove the sentences describing the aim of the study since that should be covered in the introduction
3) line 184- not sure about the word suffering. do u mean to say: 'skin or nipple areolar complex necrosis'
4) the limitation should include the relevance of the cost analysis to that particular centre since the cost calculation would vary from centre to centre/country
5) did the decrease in the length of stay for prepectoral group resulted in a catch up on the price difference of the mesh used in prepectoral group? since the overall cost incurred for each patient is more important than the cost per implant
6) though you did a predictive score for postop complications but the AUC was not impressive with AUC<0.7. not sure if you still want to include it in the paper since the predictive value is not great
7) finally, pls clearly state what additional value this paper will add to current literature since there are already similar studies available?
Comments on the Quality of English Language
Extensive edits of English language are needed.
- the title has duplicate of prepectoral and subpectoral implant, which was not neccessary
- for line 44, instead of realised, it can be replaced by done
-instead of the use of mono-centric cohort, can use single centre cohort
- line 28, instead of successive years, suggest to use in recent years
Author Response
Answer: thank you for this review and your comments
however there are some issues with this paper
- pls explain the abbreviation of LPOH in line 5
Answer: length of hospitalization (LPOH) has been delete and replaced by length of stay (LOS).
- for discussion- pls remove the sentences describing the aim of the study since that should be covered in the introduction
Answer: The sentence “The objective of this study was to compare the sub- or prepectoral implant positioning relative to complication rates, length of the surgical procedure and of hospitalization, time to adjuvant therapies, patients’ satisfaction, and procedure costs” has been delete.
- line 184- not sure about the word suffering. do u mean to say: 'skin or nipple areolar complex necrosis'
Answer: the word “suffering” has been delete and replaced by “poor blood supply” without complete necrosis: The most common complications were poor blood supply or necrosis of the skin or nipple areolar complex (45.6% of complications), hematoma (30.1%), and infection (16.2%) with no significant difference between subpectoral and prepectoral implant-M-IBR.
- the limitation should include the relevance of the cost analysis to that particular centre since the cost calculation would vary from centre to centre/country
Answer: we have added: Thirdly, the relevance of the cost analysis to the particular centre, as the cost calculation would vary from centre to centre/country.
- did the decrease in the length of stay for prepectoral group resulted in a catch up on the price difference of the mesh used in prepectoral group? since the overall cost incurred for each patient is more important than the cost per implant
Answer: In chapter Results, paragraph 3.6. Cost evaluation: median cost was 3560 Euros for an implant in the subpectoral position and then was 3305 Euros for prepectoral position without mesh and then the cost of a pre-pectoral implant procedure is 255 euros less expensive than a subpectoral implant procedure (-7.7%).
6) though you did a predictive score for postop complications but the AUC was not impressive with AUC<0.7. not sure if you still want to include it in the paper since the predictive value is not great
Answer: We agree with you: the AUC of the predictive score is low. However, the performance of this score could be improved by a larger, multicentre study, which is already planned and underway. We have added in chapter conclusion: “These results need to be confirmed in other centres, and the predictive score for complications needs to be improved, in particular by a larger multicentre study currently underway”.
7) finally, pls clearly state what additional value this paper will add to current literature since there are already similar studies available?
Answer: In chapter conclusion we have added texte lined to yellow: The results of our study provide further evidence that prepectoral implants do not lead to higher complication rates, are shorter, and less expensive, provided that mesh is not used. Furthermore, compared with previous studies, we have shown that the time to adjuvant therapy does not differ from subpectoral implants and patients’ satisfaction remains similar.
Comments on the Quality of English Language
Extensive edits of English language are needed.
- the title has duplicate of prepectoral and subpectoral implant, which was not necessary: : correction made: Postoperative outcomes of Pre-pectoral versus sub-pectoral implant immediate breast reconstruction
- for line 44, instead of realised, it can be replaced by done: correction made
-instead of the use of mono-centric cohort, can use single centre cohort: correction made
- line 28, instead of successive years, suggest to use in recent years: correction made
Round 2
Reviewer 2 Report
Comments and Suggestions for Authors
since the predictive scores have low AUC and will be validated in planned larger study, i would suggest that these scores be omitted from this study. they could be reported later in the larger study when they have been proven to be of better predictive value.
while prepectoral implants can cost more with mesh, this group also had shorter LOS. so, based on analysis by patient, did it resulted in lower cost for a patient, taking into account the LOS factor, with subpectoral or prepectoral implant?
Comments on the Quality of English LanguageQuality of English has improved
Author Response
Answer to reviewer comments, with modifications highlighted in blue in this answer and in the manuscript:
Since the predictive scores have low AUC and will be validated in planned larger study, i would suggest that these scores be omitted from this study. they could be reported later in the larger study when they have been proven to be of better predictive value.
Answer: The score results paragraph has been removed, with only a short information: Calculation of predictive scores for any complication or grade 2-3 complication, using the ORs of significant factors from the regression analysis isolated four and five risk groups, respectively. A higher score predicted a higher risk of events, but with a low AUC value (< 0.70).
In chapter Discussion (line 405), we have added the text highlighted in blue: The key issue in clinical practice is the occurrence of grade 2-3 complications, which increase significantly with the risk group, but with a low predictive value (0.678).
While prepectoral implants can cost more with mesh, this group also had shorter LOS. so, based on analysis by patient, did it resulted in lower cost for a patient, taking into account the LOS factor, with subpectoral or prepectoral implant?
Answer: we have added: Taking into account the shorter LOS for prepectoral implant, it resulted in lower costs was than the subpectoral implant and higher costs than the prepectoral implant-IBR with mesh.
